# Using topic modelling for unsupervised annotation of electronic health records to identify an outbreak of disease in UK dogs

**Peter-John Mäntylä Noble** [1]*, **Charlotte Appleton**[2], **Alan David Radford**[1], **Goran Nenadic**[3]

**1** Institute of Infection, Veterinary and Ecological Sciences, University of Liverpool, Neston, Wirral, United Kingdom, **2** Centre for Health Informatics, Computing, and Statistics (CHICAS), Lancaster Medical School, Lancaster University, Lancaster, United Kingdom, **3** Department of Computer Science, University of Manchester, Manchester, United Kingdom

* rtnorle@liverpool.ac.uk

**Data Availability Statement:** Clinical notes data cannot be shared publicly because they may contain occasional personal identifiers. Data are available from SAVSNET (http://liverpool.ac.uk/

## Abstract

A key goal of disease surveillance is to identify outbreaks of known or novel diseases in a timely manner. Such an outbreak occurred in the UK associated with acute vomiting in dogs between December 2019 and March 2020. We tracked this outbreak using the clinical free text component of anonymised electronic health records (EHRs) collected from a sentinel network of participating veterinary practices. We sourced the free text (narrative) component of each EHR supplemented with one of 10 practitioner-derived main presenting complaints (MPCs), with the 'gastroenteric' MPC identifying cases involved in the disease outbreak. Such clinician-derived annotation systems can suffer from poor compliance requiring retrospective, often manual, coding, thereby limiting real-time usability, especially where an outbreak of a novel disease might not present clinically as a currently recognised syndrome or MPC. Here, we investigate the use of an unsupervised method of EHR annotation using latent Dirichlet allocation topic-modelling to identify topics inherent within the clinical narrative component of EHRs. The model comprised 30 topics which were used to annotate EHRs spanning the natural disease outbreak and investigate whether any given topic might mirror the outbreak time-course. Narratives were annotated using the Gensim Library LdaModel module for the topic best representing the text within them. Counts for narratives labelled with one of the topics significantly matched the disease outbreak based on the practitioner-derived 'gastroenteric' MPC (Spearman correlation 0.978); no other topics showed a similar time course. Using artificially injected outbreaks, it was possible to see other topics that would match other MPCs including respiratory disease. The underlying topics were readily evaluated using simple word-cloud representations and using a freely available package (LDAVis) providing rapid insight into the clinical basis of each topic. This work clearly shows that unsupervised record annotation using topic modelling linked to simple text visualisations can provide an easily interrogable method to identify and characterise outbreaks and other anomalies of known and previously un-characterised diseases based on changes in clinical narratives.

savsnet) for researchers who meet the criteria for access to confidential data through a simple application process (https://savsnetvet.liverpool.ac.uk/DAPP/). On completing a data request researchers can specify which data they are requesting including the data from this study. The topic model used in this study is available through github: https://github.com/pjmnoble/Clinical-Narrative-Topic-Model.

**Funding:** ADR, PJN, GN No funder reference number, DogsTrust, https://www.dogstrust.org.uk ADR, PJN, GN BB/N019547/1, Biotechnology and Biological Sciences Research Council, https://bbsrc.ukri.org ADR, PJN No funder reference number, British Small Animal Veterinary Association, https://www.bsava.com The funders had no role in study design, data collection and analysis, decision to publish, or preparation of the manuscript.

**Competing interests:** The authors have declared that no competing interests exist.

## Introduction

One of the aims of veterinary surveillance is to rapidly detect the emergence of new diseases. Such emerging diseases might be expected, relatively early in an outbreak, to be reflected in changes of clinical syndromes described in veterinary clinical narratives [1] and syndromic surveillance has been demonstrated to be effective for identifying disease outbreaks in human medicine [2].

In January 2020, the authors became aware of anecdotal reports of a potential increase in the numbers of dogs attending veterinary practices with an acute vomiting syndrome [3]. Using electronic health records (EHRs) collected from a sentinel network of veterinary practices and a combination of syndromic surveillance (clinical records annotated at source as having a gastroenteric presentation by attending clinicians), text mining and subsequently microbiology, we were able to describe this outbreak and identify the potential cause as a canine enteric coronavirus [4,5].

Whilst this surveillance approach provided a means to identify this outbreak, it was largely predicated on records having been previously annotated by the attending clinician for one of ten pre-determined syndromes with subsequent analysis of selected data. Other classification systems exist for annotation of veterinary clinical notes covering a much wider range of diagnoses and syndromes [6,7] but may also be incompletely used at point of care requiring additional search methods (such as key word matching) and manual reading to detect syndrome occurrence [8], restricting their utility for real-time surveillance. Additionally, regardless of who applies them, clinical coding systems may fall victim to inaccurate, inconsistent and incomplete coding by users [9,10].

Methods to circumvent the need for manual clinical coding that are based on pattern-matching for specific key words and combinations of key words present in free-text narratives require careful management of and frequent phenomena of abbreviation, misspelling and negation [11,12]. Additionally, these approaches can require a prediction of all the potential ways in which an emerging disease might be represented in the clinical records before they occur. Whilst various supervised machine-learning approaches have been applied to automate EHR annotation with syndromically-relevant classifications, these typically require pre-annotated data for model training [13–16] which requires considerable investment of time by domain experts to annotate records.

Developing supervised annotation systems to detect outbreaks can also be challenging where data from actual disease outbreaks is not available or where a new syndrome has emerged. Veterinary examples addressing this have included injecting datasets with virtual cases based on expert-derived clinical criteria [17] or based on feature extraction from known cases [18].

A system that allows truly unsupervised generation of annotations of EHRs based on clinical narrative content could allow detection of outbreaks of both known and novel syndromes in a timelier manner than rule based and trained machine-learning solutions.

Topic modelling is an unsupervised approach that assumes collections of documents can be categorised as having content with a probability of relating to specific topics where each topic is represented by the probability of any given word from the full corpus of text being present within that topic [19]. This method has been evaluated for identification of disease outbreaks in news media, clinical disease staging, medication prescribing patterns and adverse drug reaction classification [20–23] and in classifying EHRs for identification of phenotype and some clinical syndromes in human medicine [24,25]. Here, for the first time, we apply topic modelling to a large corpus of veterinary clinical narratives and evaluate whether it would have detected a gastroenteric disease outbreak using data retrospectively collected during this rare

event. We also model its utility to detect outbreaks of other syndromes using a novel method of injecting practitioner-defined genuine clinical narratives into existing health data.

## Methods

### Data collection

The Small Animal Veterinary Surveillance Network (SAVSNET) was established in 2010 to collect electronic health records (EHRs) at scale from a network of sentinel veterinary practices across the UK [26]. Each EHR includes a consultation narrative as well as de-identified patient information (e.g. age, sex, species, breed, weight, vaccination status, drugs and treatments pre-scribed) from over 500 contributing premises in the UK in real-time. In addition, the attend-ing veterinary clinician must add a single syndromic annotation, termed main presenting complaint (MPC) to each EHR, selecting from a short list comprising: gastroenteric, respira-tory, pruritus, tumour, kidney disease, other unwell, post-op check, vaccination and other healthy, using a simple user interface presented at the end of each consultation (Fig 1, [27]). The proportions of each of these MPCs are monitored in real-time and contribute to regular surveillance reports [28,29]. SAVSNET has ethical approval to collect and use these data from the University of Liverpool ethics committee (RETH000964).

We analysed a corpus of SAVSNET clinical narratives for dogs only. Records were stored and manipulated using Python [30] Django (Django Software Foundation) and MySQL [31].

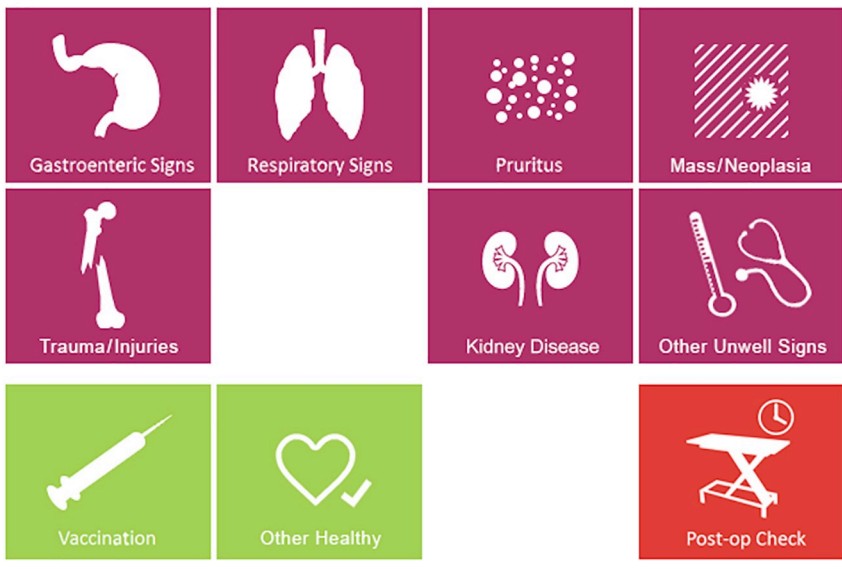

**Fig 1. SAVSNET MPC window.** At the end of each consultation, veterinary clinicians are required to annotate the health record selecting from a list of syndromes or main presenting complaints (MPCs) as shown.

Datasets were imported as Pandas dataframes [32]. Text preparation utilized the natural language tool kit (NLTK) suite [33].

## Topic model development

Latent Dirichlet allocation (LDA) topic modelling and collocation analysis (n-gram discovery) were performed using the Gensim library [34].

A training set of 200,000 narratives from EHRs dated between 1.3.2019 and 1.10.2019 was selected from the corpus. These narratives were tokenised at white space and non-word character boundaries and part-of speech tagged using NLTK, retaining noun, verb, adjective and number tokens. Common word collocations (bigrams, trigrams and quadgrams) were identified using the Gensim Phrases method and narratives were reduced to 'bag of words' documents. The Gensim Latent Dirichlet Model class can accept a topic count as a parameter prior to creating the topic model. This topic count was chosen on the basis of evaluating the 'u_mass' coherence property of the Gensim Coherence class using topic models created using between 2 and 100 topics.

The best suited topic count was used to generate a topic model which was used to analyse all narratives in the corpus after 1st January 2016 (a time point after which the number of veterinary practices participating in SAVSNET was comparatively stable). The probability of each topic was estimated for each narrative. For surveillance-testing, narratives were finally annotated with a single topic where the model assigned a probability of over 0.5 for that topic (i.e. that topic was more likely than any of the others combined). Where no topics exceeded a probability of 0.5 for a narrative, the narrative was annotated as 'not classified'. Topic counts were assessed to evaluate their degree of concordance with existing MPCs as recorded by participating practitioners. An adjusted mutual information score was calculated for each topic with each MPC using the SciKit Learn using the adjusted_mutual_info_score metric [35]. All model creation was performed on a quad core Xenon processor with hyperthreading in a Dell Z440 workstation (Dell Computers inc) equipped with 64Gb RAM.

## Simulated outbreak

To evaluate the potential for the final model to detect outbreaks of other given syndromes, six simulated outbreaks lasting 60 days were created using existing narratives from EHR labelled by veterinary surgeons as having specific MPCs (gastrointestinal, respiratory, pruritus, trauma, tumour and kidney disease). For each day of each simulated outbreak a sample of 100 EHRs (the scale of the increase in daily consult counts typical for the natural disease outbreak) with a given MPC was randomly selected from all dog consultations in the full SAVSNET database. These samples were inserted into a copy of the full database with timestamps adjusted to fit within the 60-day simulated outbreak for that MPC. All records from these simulated outbreaks database were then annotated using the topic model as described above and MPC counts were then visually compared with record counts for each topic.

## Gastroenteric disease outbreak

Daily counts for clinical narratives annotated with the gastroenteric MPC were plotted as 7-day rolling averages and visually compared to narrative counts annotated with topics from the topic model. Daily narrative counts labelled with gastroenteric MPC and each topic were compared with Spearman's rank correlation both for one year preceding and during the outbreak period.

Data for the topic best matching the outbreak was plotted with trend lines calculated using a Bayesian binomial generalised linear model trained on weekly prevalence between 2014 and

2019 [29], allowing us to identify observations that were extremely (>99% credible intervals) or moderately (>95% credible intervals) likely to be outside normal variation.

### Topic interpretation

In order to infer topic meaning, topic word-content was evaluated using word clouds generated using the WordCloud module in python [36]. For a more quantitative approach, we used LDAvis [37]. This method allows visualisation of words occurring with high probability in the topic, weighting for words that do not appear frequently in other topics by setting a relevance metric. We used a relevance metric of 0.2 allowing visualisation of topic key words found more commonly in the topic than in the general corpus.

### Dynamic topic modelling

Evolution of the topic model in terms of token-weightings for each topic over time was assessed using dynamic topic modelling [38]. A sample of 120,000 narratives from dog consultations between 1.3.2019 and 1.3.2020 (the 12-month period encompassing the outbreak) were taken and processed using the Gensim LdaSeqModel class, seeding the model with the original LDA model and time slicing at two-month intervals (approximately 20,000 records per time slice). Word weighting for the topic best matching gastroenteric disease was reviewed.

## Results

### Topic modelling

When evaluating the most appropriate topic count to use for this study, topic coherence score improved notably between 2 and 30 topics (S1 Fig). Topic model production took 1.25, 1.75,3.6, 8.5 and 22 hours to produce models with 10,30,50,70 and 90 topic models respectively. We therefore decided to perform experiments with 30 topics.

Using the 30-topic model developed on 200,000 records to annotate the 3,499,566 records in the dataset, only 982,333 (28.1%) were allocated a primary topic (where its topic probability exceeded 0.5) with individual topics labelling between 46 (0.001%) and 114034 (3.258%) narratives (S2 Fig) When narratives grouped by MPC-annotation were assessed for the primary topics with which they were labelled, gastroenteric, respiratory, trauma and post-op MPCs had the four largest majority topics by proportion of labelled topics (topic_17, topic_6, topic_26 and topic_13 respectively; Fig 2A). Other MPCs appeared to comprise two or three main contributors (e.g. pruritis: topic_22 and topic_18, tumour: topic_20, topic_9). Analysis of adjusted mutual information clustering confirmed association between topic_17 and gastroenteric MPC as well as topic_6 and respiratory mpc, topic_13 and post-operative checks, topic 26 and trauma, topic_9 and tumour (Fig 2B).

### Simulated outbreaks

For simulated outbreaks, increased cases were clearly visible for the relevant MPCs but only elicited visibly detectable signals for topic_17 (correlating with injected gastroenteric narratives and topic_6 correlating with injected respiratory narratives (Fig 3).

### Gastroenteric disease outbreak

Only topic_17 counts showed a similar rise to gastroenteric MPC counts during the period of the outbreak (Fig 4A). To compare overall time-series of data, time-series of more common topics (those with over 25 narratives/day), normalised to their average daily count were plotted along with the time-series for MPCs. The outbreak of GI disease as identified by the increase

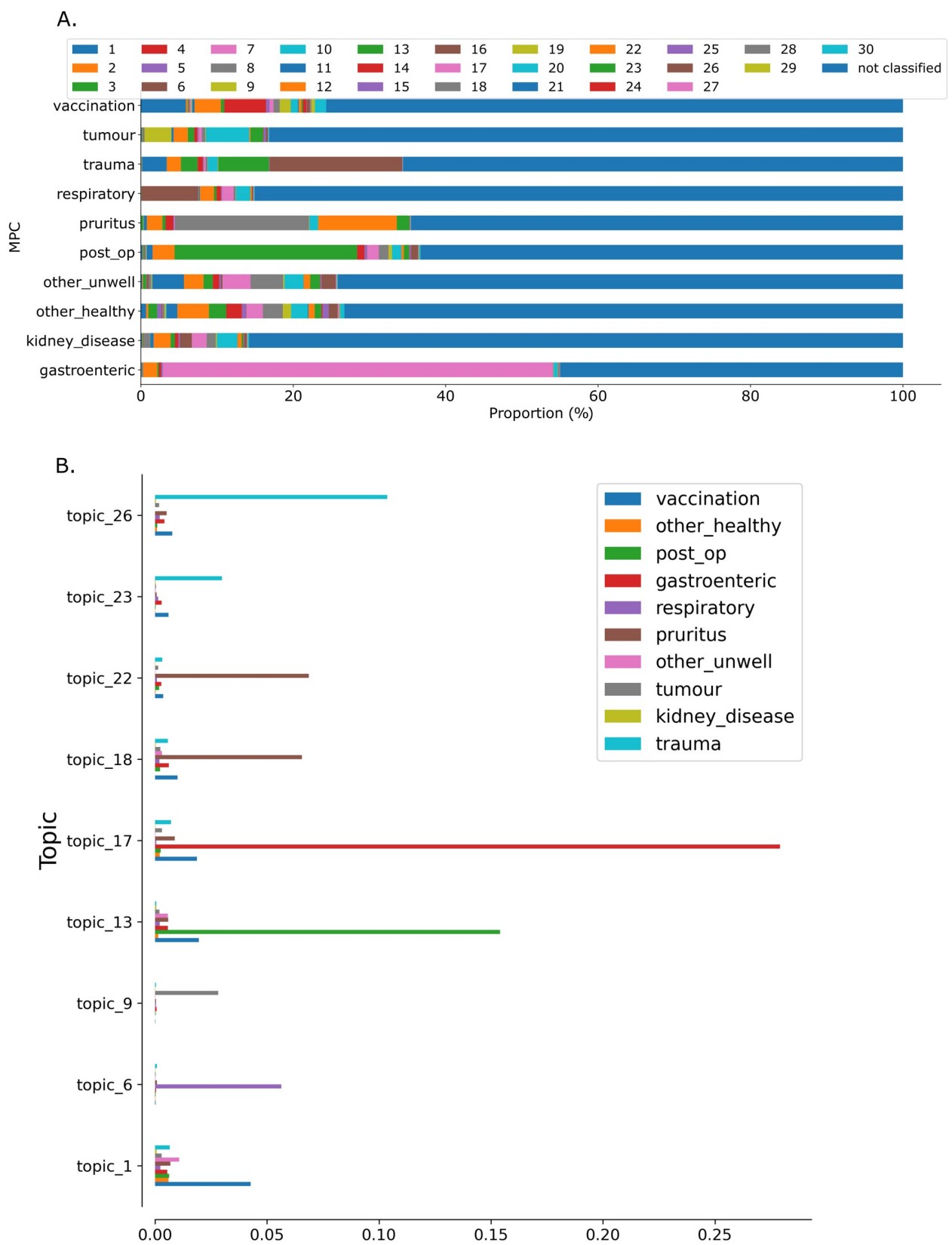

**Fig 2. Count and topic distribution for different MPCs amongst 3,499,566 veterinary electronic health records.** (A) Proportion of narratives belonging to each MPC classified as relating to each topic. (B) Adjusted mutual information score for topics with the indicated MPCs showing topics where that had at least one adjusted mutual information score of greater tan 0.025.

in gastroenteric MPC reporting was closely paralleled by the topic_17 count but by no other topics (Fig 4B).

Daily narrative counts matching topic_17 and gastroenteric MPC were distributed similarly with a Spearman correlation of 0.927(95%CI: 0.911–0.940) prior to the outbreak (Fig 5A) which rose to 0.978 (95%CI: 0.968–0.985) during the outbreak (1/11/2019-1/3/2020; Fig 5B) where correlation with other topics during that period ranged from 0.305–0.766. When plotted with trend lines calculated using a Bayesian binomial generalised linear model as described above, a clear series of weeks were highlighted where counts for topic_17-classified narratives were outside the 99% credibility limit in December-February, matching the pattern seen for gastroenteric MPC reporting. (Fig 5B and 5C).

### Topic interpretation

Based on word-cloud assessment, key words contributing to topic_17 included series of words describing gastrointestinal function including gastrointestinal clinical signs and their abbreviations ('diarrhoea', 'vomiting', 'd+' and 'v+') as well as words relating to gastrointestinal functions ('eating', 'faeces', 'stools') and words relating to time intervals ('yesterday', 'morning') (Fig 6A). Using LDAVis revealed some more subtle detail identifying words including 'eaten, 'ate', as well as words describing common treatments for gastrointestinal disease ('bland diet' and 'chicken' and 'omeprazole') and other clinical signs ('lethargy') (Fig 6B).

Topic_6 was represented by words relating to cough and heart disease and topics 18 and 22, which were highly represented in pruritus MPC cases were characterised by words relating to ear problems (topic 18) and allergic skin disease (topic_22; Fig 6A).

Several topics appeared to describe common syndromes in veterinary practice that did not have an associated MPC such as anal gland disease (topic_3), nail-overgrowth (topic_5), reproductive disease (topic_10), ocular disease (topic_11), body weight issues (topic_21), lameness (topic_26), dental disease (topic_28). In addition, there were topics that appeared to describe preventative health visits (topic_1) and routine clinical examination findings (topic_24). There were some topics that were not immediately interpretable (topic_12, topic_15).

### Dynamic topic modelling

Topic evolution during the outbreak period, evaluated using dynamic topic modelling seeded with the initial topic model revealed an increasing priority of words relating to vomiting ('vomiting', 'v+') within the gastroenteric topic (Fig 7). Other words maintained a similar weighting before and during the outbreak.

### Discussion

Topic modelling has been used in the medical literature to identify syndromic features of clinical records [23], track disease prevalence in social media [39,40] and annotate human clinical notes datasets to evaluate potential genotype correlation with topics or to classify for ICD-10 annotation [24,25]. Here, for the first time we demonstrate its utility for rapid and unsupervised disease outbreak detection using the clinical narrative component of veterinary EHRs and show how simple deconstructions of the word foundations for several individual topics

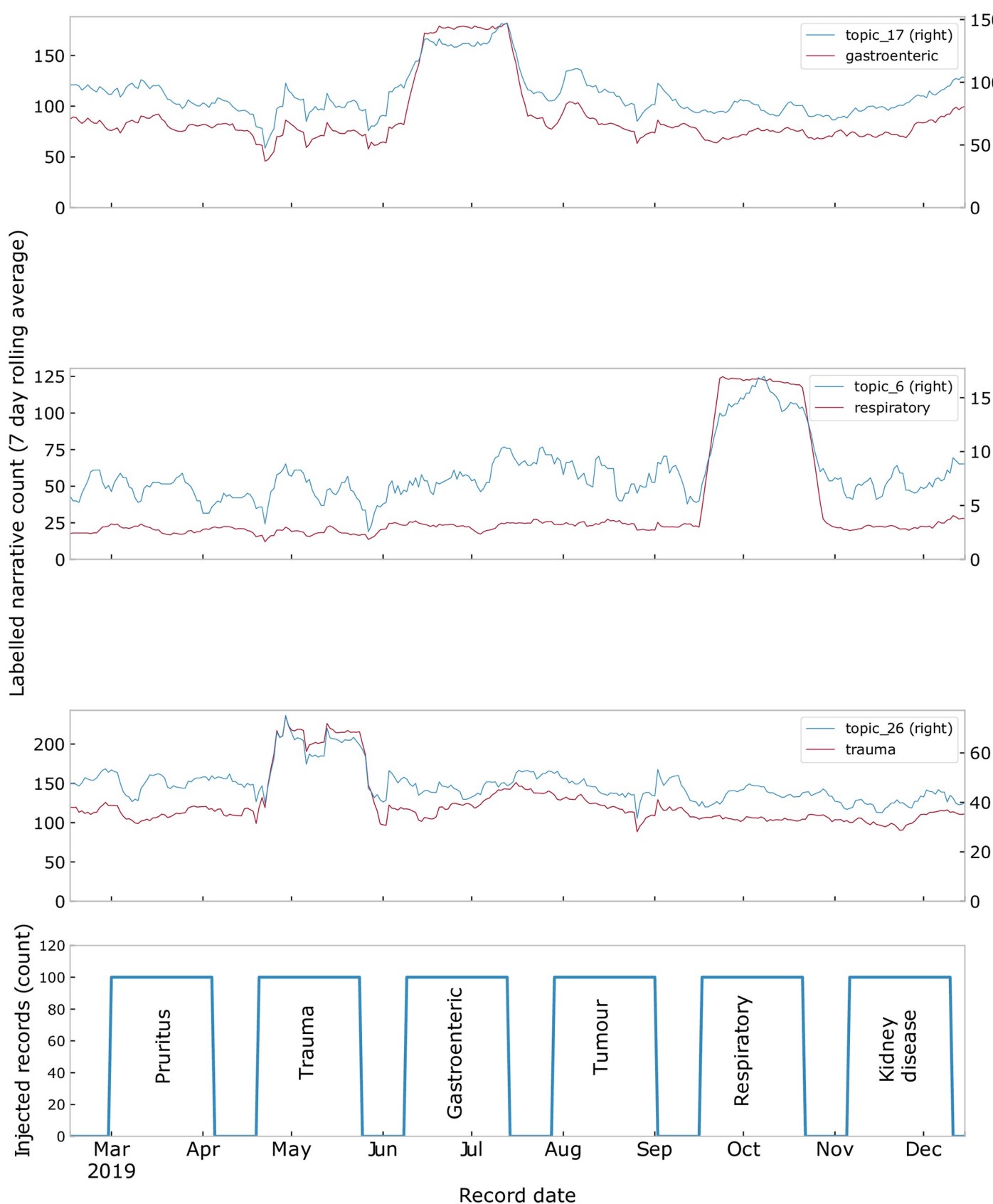

**Fig 3. The effect of injected outbreaks on topic counts.** Simulated outbreaks were injected for pruritus, trauma, gastroenteric, tumour, respiratory and kidney disease with signals visible for gastroenteric MPC (topic_17), respiratory MPV (topic_6) and trauma MPC (topic 26).

can provide transparent, clinician-friendly outputs to understand the significance of any signals.

On assessment of records annotated by clinicians for a limited set of MPCs, it was clear that specific MPCS were associated with varied distributions of topics. In the case of the gastroenteric MPC, topic_17 was the majority topic but other MPCs had also majority topics, thus respiratory, trauma and post-op MPCs were, in the majority annotated with topic_6, topic_26 and topic_13 respectively.

Clearly the criterion for apportioning a topic were set rather rigorously leading to many narratives remaining unclassified with respects to topic. Additionally, most narratives were associated with additional topics, albeit with lower probabilities than that of the one with which was their majority topic. Considering these lower probability topics may well provide additional or complementary signals regarding presentation for multiple syndromes and will be evaluated in further studies. That said, even under the conditions set in the current study, the gastroenteric MPC was matched by topic_17, with most of the remaining being unclassified, suggesting a very distinct terminology used for this subset of gastroenteric MPC narratives. This is perhaps not surprising given that gastroenteric disease is most frequently associated with a limited combination of clinical signs (vomiting and diarrhoea). The limited cover of some MPCs by single majority topics may reflect the diversity of clinical syndromes associated with the MPC. For instance, with tumour MPC, varied clinical presentations (and presumably topics) would reflect the variety of sites affected by tumours with some common features (e.g., masses) reflected in one of the topics only (topic_9). Likewise, the 'other unwell' MPC was inevitable to comprise many varied clinical syndromes and thus different topics.

Only topic_17 showed any evidence of a change in daily record counts when records were screened over the period of the natural outbreak with daily topic_17 counts correlating significantly with gastroenteric MPC counts. Furthermore, topic_17 narrative count correlated more strongly with gastroenteric MPC count during the outbreak implying topic_17 was a closer match for narratives describing the outbreak-syndrome than for normal background gastroenteric disease. The smaller positive correlation identified between gastroenteric MPC and other topics probably reflected weekly variation of consultation counts (few on Saturdays and even fewer on Sundays) affecting all topics and MPCs.

It was critical to see whether an outbreak-detection model would identify the outbreak in topic-annotated records as well as it could in MPC-annotated records. Using a Bayesian binomial generalised linear model previously used to identify outbreak-candidates [29] we were able to demonstrate that the topic_17 annotated record count would have generated an alert in which counts exceeded a credibility threshold of 99% in a very similar way to the clinician-derived MPC annotations, thereby providing further evidence that topic modelling could provide a route to identifying similar outbreak in the absence of clinician-based annotation.

Our observations with *in-silico* outbreaks, suggest this was not unique to gastroenteric disease (topic 17), but that outbreaks of respiratory (topic 6) and trauma/lameness (topic 26) might also be detectable with such a system. However, not all simulated MPC outbreaks were detected. For example, pruritus appeared to have two major topic annotations (topic 18 and topic_22) there was no obvious change in narrative counts for the injected records. This probably reflects the fact that pruritus narratives form a larger proportion of the SAVSNET corpus, were divided between two topics and have a marked seasonality peaking in spring/summer [29] which would combine to make changes in the individual topics proportionally smaller.

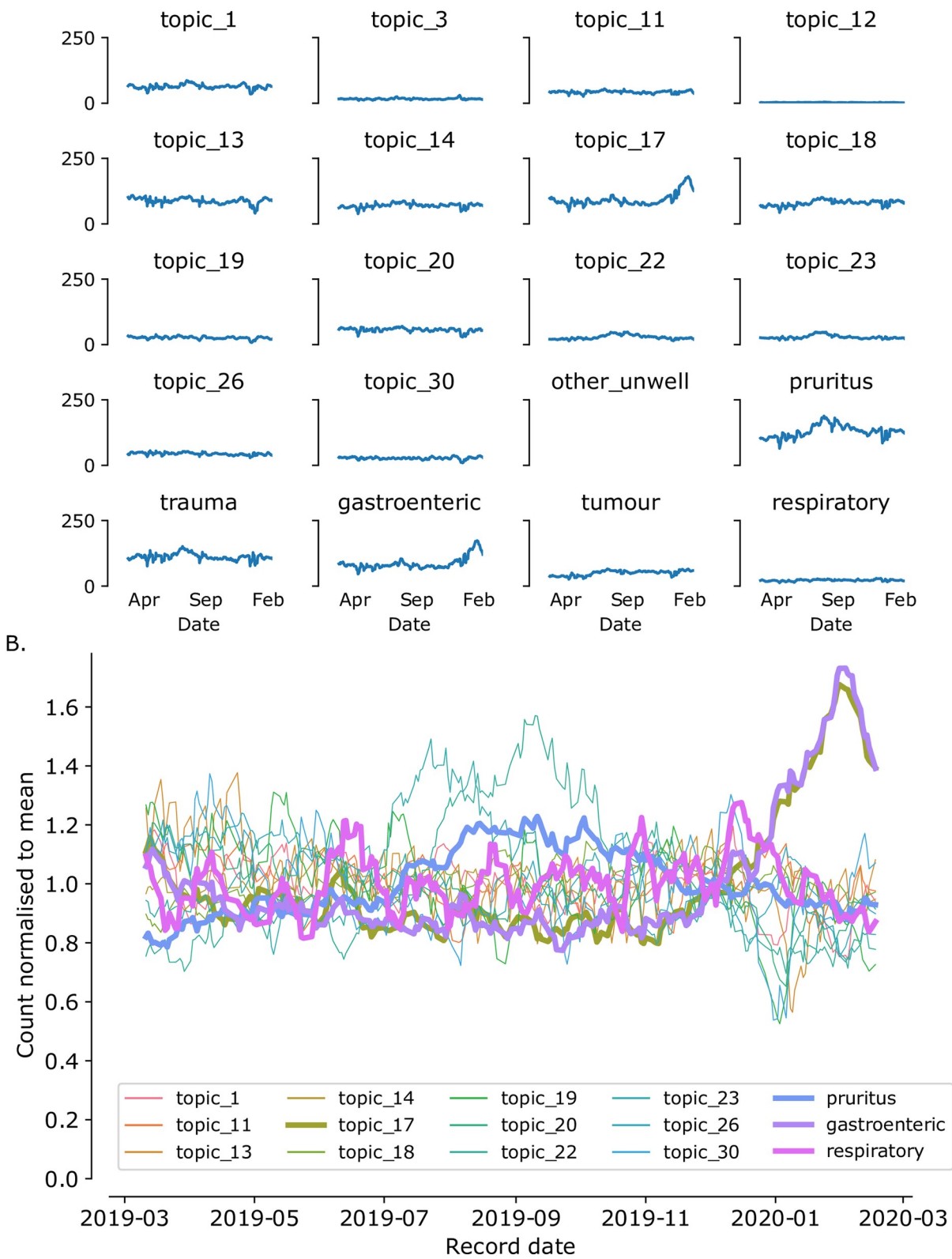

**Fig 4. Comparison of topic counts vs MPC counts over the course of the natural disease outbreak (November 2019 to March 2020).** (A) Daily count of narratives matching individual topics (where average daily consult count exceeds 10) were plotted as rolling 7-day average. (B) Counts for narratives annotated by topics and selected MPCs, normalised to the average count for that topic, plotted as rolling 7-day average. Only topic 16 counts match the gastroenteric MPC over the course of the natural disease outbreak.

Under these circumstances, injecting only pruritus MPC-labelled narratives matching selected key words might have been both more authentic in representing a syndromic outbreak and more readily detected through the topic model.

Unsupervised machine learning algorithms have been investigated to classify text and identify changes in the structure and content of text documents with time [39] and to reduce data dimensionality for subsequent analysis in other machine-learning models (reviewed in [41]). In the medical field it is critical that reason for annotation of any given record can be understood. Machine learning approaches can lead to 'black box' situations where the link between document and classification is opaque necessitating development of architectures to unravel and visualise the features underpinning a classification [41,42]. Critically, any unsupervised system used for disease surveillance needs to be interrogable in order to translate the signal into a description of the clinical syndrome underlying it that is actionable by medical practitioners, not just computer scientists. Topic modelling provides an excellent subject for this interrogation. At its simplest level, a topic can be evaluated for the most common words in the text corpus that contribute to that topic as illustrated here by word-cloud representations of the topics. Thus, topic_17 for which an outbreak could be clearly seen was instantly recognisable as describing gastroenteric disease due to key words and abbreviations relating to specific clinical signs, gastroenteric function and features relating to time all of which corroborated the observed description of cases from the actual outbreak (vomiting with or without diarrhoea occurring over a short time). The selection of more topic-specific terms using the relevance metric available through LDAVis also identified mentions of therapeutic choices common in gastroenteric disease (use of bland diet or chicken, medication with omeprazole or maropitant) and other clinical signs (lethargy) all of which were reported as key components of the outbreak syndrome. It is of note that changes in national prescribing habits for maropitant and omeprazole were key measurable features in the disease outbreak [4].

Whilst a signal was visible in the data both at the level of MPC classification and topic_17 annotation, the threshold at which an outbreak might be officially classified as such remains a challenging one. For example, had an arbitrary threshold of two or more weeks of datapoints outside the 99% credibility limitation been set as an alert threshold, the outbreak would have been automatically flagged in late December–two months before its peak. Kass *et al* developed robust methodologies to detect simulated disease outbreaks in companion animal data which use a more objective measure to signal anomalies and might be applicable to this data [43].

Here we used a topic model developed on a 200,000-record subset of narratives to evaluate the SAVSNET corpus. The topic model developed, however, may be applicable to other similar corpora of veterinary clinical notes. The use of topic models across corpora has been evaluated for instance in investigating consumer injury reports [44]. Furthermore, while the topic model presented here appeared to identify one clinical syndrome, namely gastroenteric disease, extremely well, there are a variety of ways this approach might be enhanced to improve specificity for other syndromes. For instance, a topic model might be prepared using only narratives tagged for a specific MPC or including specific keywords in order to improve selectivity for a given syndrome. Similarly, topic models have been applied to records pre-filtered using machine learning [40] in order to identify trends in twitter data regarding seasonal influenza.

The choice to model 30 topics was a pragmatic one based on a compromise between model coherence and model training time. With increasingly affordable computer power, it will be

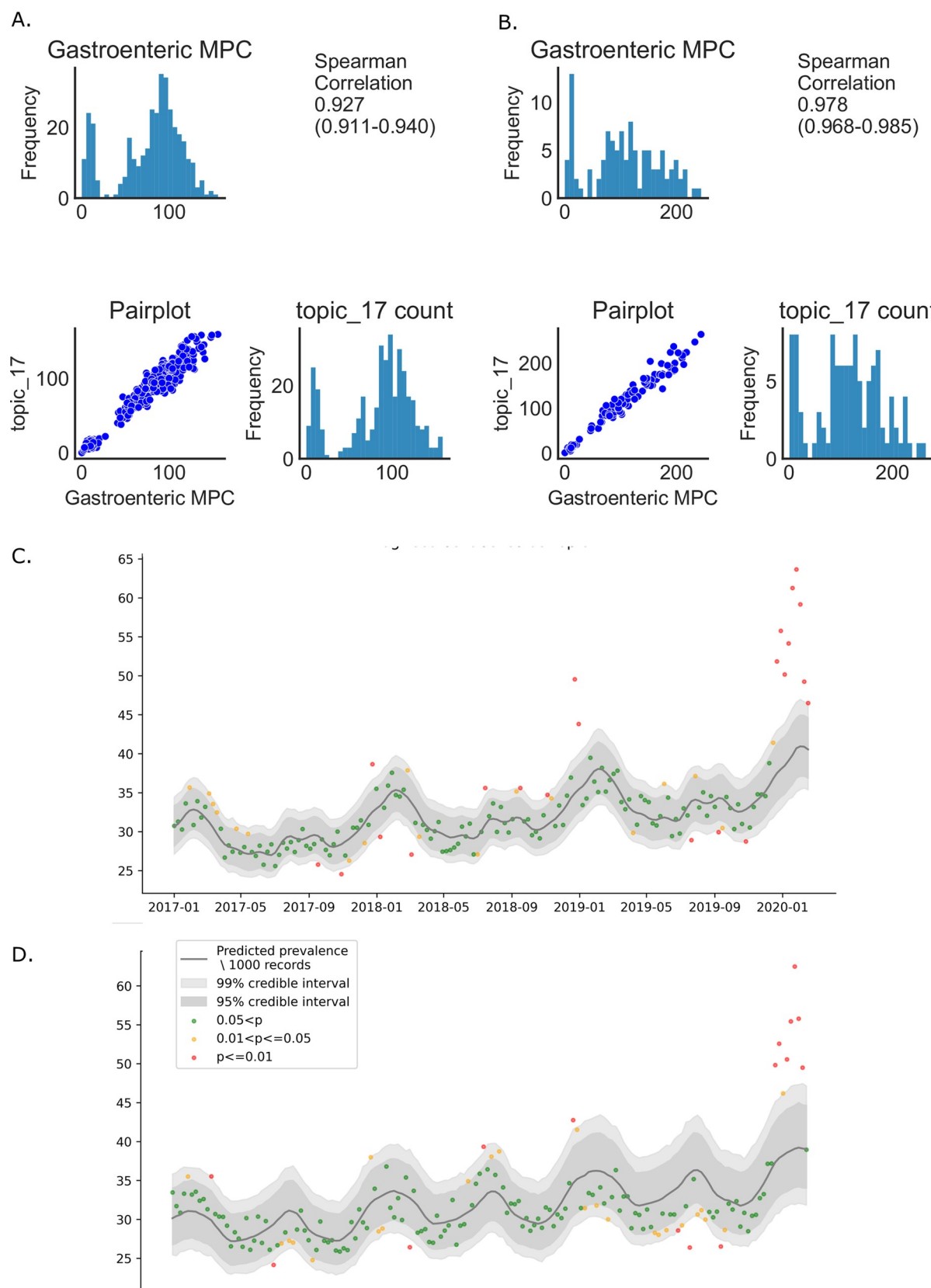

**Fig 5. Comparison of topic17 and gastroenteric MPC narrative counts.** (A) Comparison of frequency distributions of topic 17 narratives and gastroenteric MPC narratives in the year preceding the outbreak. (B) Comparison of frequency distributions of topic 17 narratives and gastroenteric MPC narratives during the outbreak. (C) and (D) Temporal pattern of topic 17 (C) and gastroenteric MPC (D) narrative counts. Red points represent the extreme outliers (outside the 99 per cent credible interval [CI]), orange points represent the moderate outliers (outside the 95 per cent CI but within the 99 per cent CI), and green points represent the average trend (within the 95 per cent CI).

easier to model at higher topic numbers which would be expected to identify more subtle features. Even using 30 topics, the model identified subsets within the pruritus-MPC records, namely topic 18 and topic 22 which word-cloud assessment identified as allergic skin disease and ear disease narratives, each distinct, clinically relevant syndromes.

In the case of respiratory disease, although topic_6 was the most common annotation for records annotated as respiratory by clinicians, key words for this topic included 'heart' and 'murmur' as well as respiratory key words such as 'coughing', 'sneezing', 'chest' and 'panting'. This overlap may reflect a tendency for uncertainty in primary veterinary consultations as to whether animals presenting with what they classify as a respiratory syndrome are primarily the result of a respiratory disease or a sequel to cardiac disease where cough is considered to be a clinical sign [45].

Word-cloud analysis identified several topics that related to common syndromes not included in the MPC list (anal gland disease, nail-overgrowth, reproductive disease, ocular disease, body weight issues, lameness). This highlights the utility of topic modelling for potentially monitoring occurrence of a wider portfolio of common syndromes affecting companion animals without requiring clinicians to make more complex coding decisions.

There were clearly topics within the model that were not immediately interpretable from their word-weightings. These may have reflected production of so called 'vacuous topics' due to requesting too many topics through the underlying statistical model. Given the evaluation of topic content, it would be possible to ignore any outbreak-like signal from these, but it would be better to prevent production of such vacuous topics through a similarly unsupervised methodology. Unsupervised enhancements to LDA model generation have been proposed [46] and will be evaluated in further studies.

One risk of this study is that using a topic model developed at a single time point and applied over a longer time-course might detect changes in language-use (for instance if a new guideline for reporting were taken up across the network) rather than emergence of a new syndrome. Dynamic topic modelling evaluates topic evolution over time. In this study, dynamic topic modelling exposed some insight into the changing priority of key words describing cases matching topic 17, namely the increased weighting for words describing vomiting which was the key presenting sign of animals experiencing the outbreak disease. In a prospective outbreak study, this evolution would allow researchers investigating an outbreak to apply more focussed text mining tools to identify specific cases with those clinical signs.

## Conclusion

Based on the current study we propose that topic modelling could be used to automatically annotate datasets of clinical records into topics, without the need for domain expert-based coding. Monitoring these topics temporally and spatially could allow detection of an anomaly or outbreak associated with a specific set of clinical signs not previously defined by the user, and as such may complement such clinician-derived annotation. On detection of a potential outbreak, the clinical language describing the clinical syndrome can be readily interpreted by those charged with controlling individual animal and population health, breaking the potential barrier that can exist with other unsupervised methods between those coding a solution and those charged with interpreting and acting on any signals. This system is computationally

A.

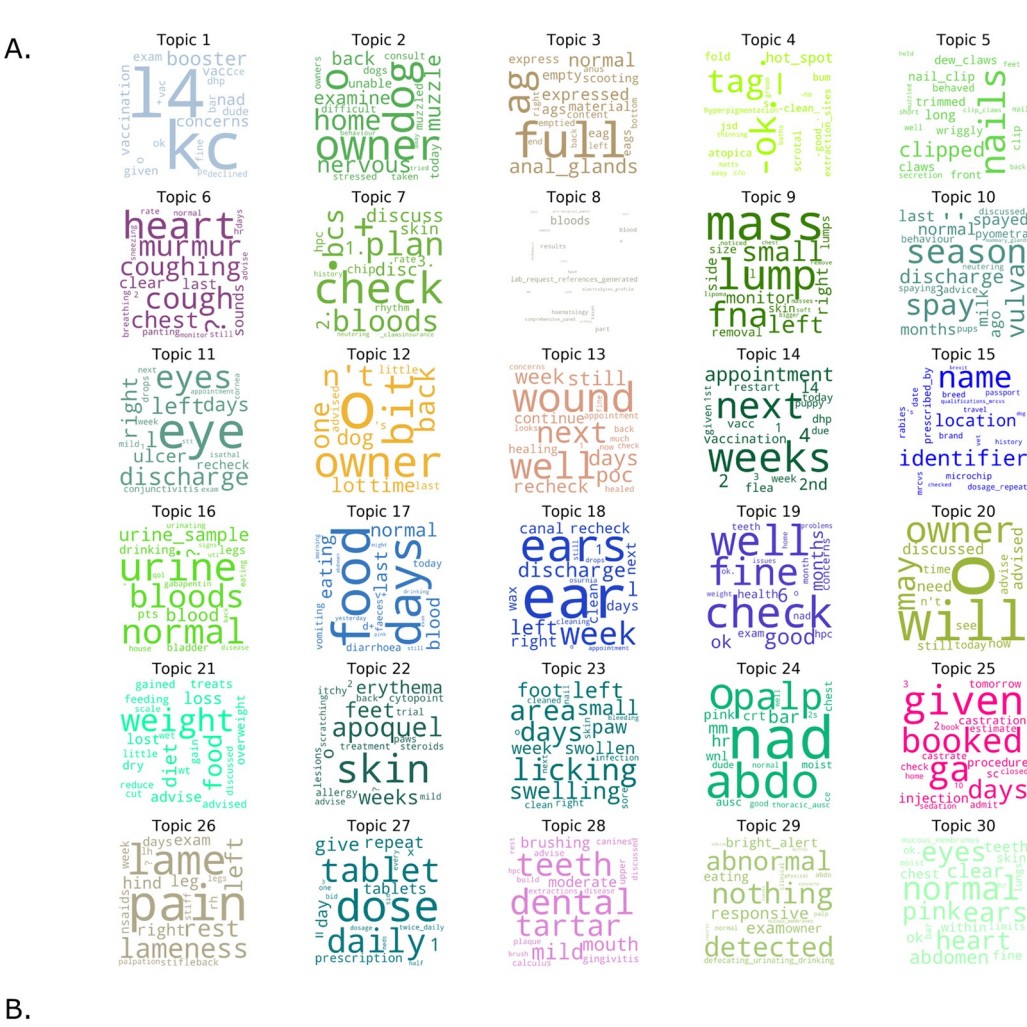

B.

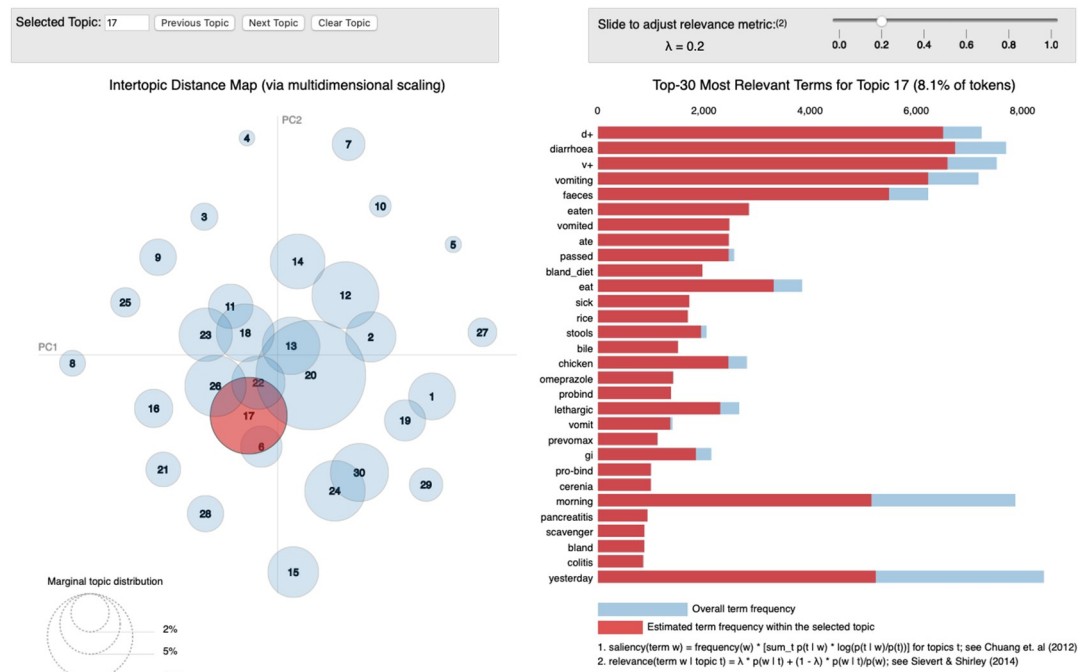

**Fig 6. Analysis of topic content.** (A) Topic representations as word clouds. (B) Analysis using LDAVis of topic separation and frequency within the corpus (left panel) and the key terms contributing to the topic 17 narrative identifying words and abbreviations related to gastrointestinal function (right panel).

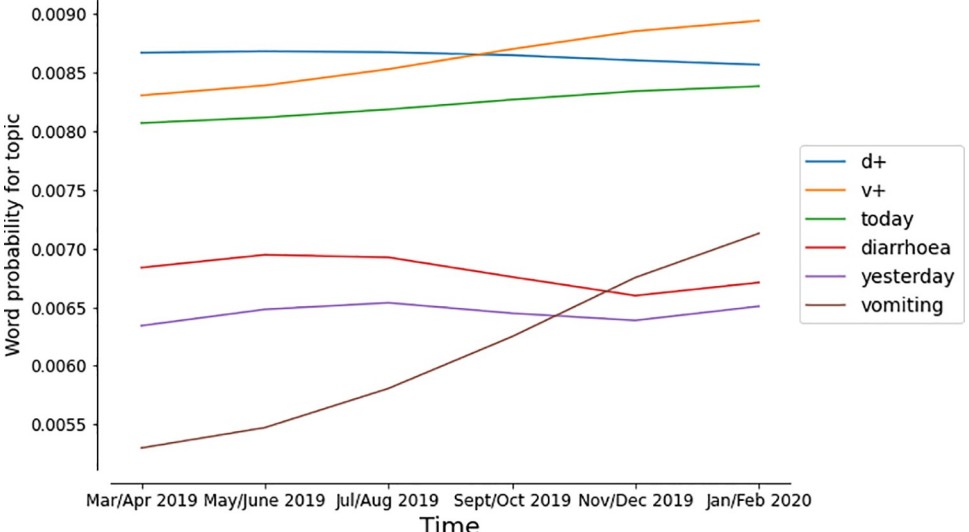

**Fig 7. Dynamic topic modelling.** The evolution of token-weighting within the gastroenteric topic showed increasing weighting for vomiting-related terms.

efficient and can run (and is running) in near real-time. Additionally, either using continuous dynamic topic modelling or repeated static modelling this method may have utility in identifying outbreaks of novel disease associated with novel syndromes without prior knowledge of what those clinical signs are.

## Supporting information

**S1 Fig. Model coherence with number of topics.**
(TIF)

**S2 Fig. Narrative counts for each topic.**
(TIF)

**S1 File. Clinical record data annotated with MPC and topic number.**
(CSV)

**S2 File. Clinical record data with injected outbreaks annotated with MPC and topic number.**
(CSV)

## Acknowledgments

We wish to thank data providers in veterinary practice (VetSolutions, Teleos, CVS and independent practitioners) without whose participation this research would not be possible. Finally, we are especially grateful for the help and support provided by SAVSNET team members Beth Brant, Susan Bolan and Steven Smyth.

## Author Contributions

**Conceptualization:** Peter-John Mäntylä Noble, Alan David Radford, Goran Nenadic.

**Data curation:** Peter-John Mäntylä Noble, Charlotte Appleton.

**Formal analysis:** Peter-John Mäntylä Noble, Charlotte Appleton.

**Methodology:** Peter-John Mäntylä Noble, Charlotte Appleton.

**Project administration:** Peter-John Mäntylä Noble, Alan David Radford.

**Software:** Peter-John Mäntylä Noble.

**Supervision:** Alan David Radford, Goran Nenadic.

**Visualization:** Peter-John Mäntylä Noble, Charlotte Appleton.

**Writing – original draft:** Peter-John Mäntylä Noble.

**Writing – review & editing:** Peter-John Mäntylä Noble, Charlotte Appleton, Alan David Radford, Goran Nenadic.

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
