## [Decision Letter · Decision Letter 0]

27 Apr 2021

PONE-D-21-01941

Using topic modelling for unsupervised annotation of electronic health records to identify an outbreak of disease in UK dogs.

PLOS ONE

Dear Dr. Noble,

Thank you for submitting your manuscript to PLOS ONE, and for your continued patience. I am aware that this first round of review took longer than usual, but I trust that you and your co-authors understand the current difficulties in finding researchers with availability to contribute their time to the peer-review process.

I am happy to report that we found two experience reviewers, who after careful consideration, suggested moderate changes, as pasted below. Therefore, we invite you to submit a revised version of the manuscript that addresses the points raised during the review process.

We look forward to receiving your revised manuscript.

Kind regards,

Fernanda C. Dórea

Academic Editor

PLOS ONE

Journal Requirements:

2. We noted in your submission details that a portion of your manuscript may have been presented or published elsewhere.

[Figure 6c is very similar to a figure in a paper submitted for publication (Radford et al 2021). In the current paper, this panel shows the trajectory of a disease outbreak which is described in the Radford et al paper as a comparison to the signal detected by topic modelling which is under discussion in the current paper. The Radford paper is primarily reporting the disease outbreak. In the current paper, we are illustrating how a novel method of automated record annotation could have highlighted the outbreak report in the Radford et al paper and feel that its inclusion clearly illustrates the utility of topic modelling whilst not in any way recapitulating the findings of the Radford et al study.]

[The data for this project was supported by generous fundingfrom BBSRC,DogsTrustas part of SAVSNET-Agile,andpreviouslyBSAVA. We wish to thank data providers in veterinary practice (VetSolutions, Teleos, CVS andindependentpractitioners)without whose support and participation this research would not be possible. Finally, we are especially grateful for the help and support provided by **SAVSNET** team members Beth Brant, Susan Bolanand Steven Smyth.]

 [ADR, PJN, GN  No funder reference number, DogsTrust, https://www.dogstrust.org.uk

ADR, PJN, GN BB/N019547/1, Biotechnology and Biological Sciences Research Council, https://bbsrc.ukri.org

ADR, PJN No funder reference number, British Small Animal Veterinary Association, https://www.bsava.com

The funders had no role in study design, data collection and analysis, decision to publish, or preparation of the manuscript.]

Additionally, because some of your funding information pertains to [commercial funding//patents], we ask you to provide an updated Competing Interests statement, declaring all sources of commercial funding.

In your Competing Interests statement, please confirm that your commercial funding does not alter your adherence to PLOS ONE Editorial policies and criteria by including the following statement: "This does not alter our adherence to PLOS ONE policies on sharing data and materials.” as detailed online in our guide for authors  http://journals.plos.org/plosone/s/competing-interests.  If this statement is not true and your adherence to PLOS policies on sharing data and materials is altered, please explain how.

Please include the updated Competing Interests Statement and Funding Statement in your cover letter. We will change the online submission form on your behalf.

Reviewers' comments:

Reviewer's Responses to Questions

**Comments to the Author**

1. Is the manuscript technically sound, and do the data support the conclusions?

Reviewer #1: Yes

Reviewer #2: Partly

2. Has the statistical analysis been performed appropriately and rigorously? 

Reviewer #1: Yes

Reviewer #2: I Don't Know

3. Have the authors made all data underlying the findings in their manuscript fully available?

Reviewer #1: Yes

Reviewer #2: Yes

4. Is the manuscript presented in an intelligible fashion and written in standard English?

Reviewer #1: Yes

Reviewer #2: Yes

5. Review Comments to the Author

Reviewer #1: The authors used topic model to mine text data to mine veterinary clinical narratives. They find that one of the topic mixture probabilities (topic 17) over time are indicative of gastroenteric presenting complaint (MPC). Using Bayesian binomial linear model, they confirm that the finding is statistically significant. The study show that it is possible to use topic model to detect outbreak of novel disease. Although there is no methodological innovations in topic modeling and the analyses are standard topic model analysis, it is a novel application and an interesting angle that the authors took in looking into the outbreak using topic models.

Major comments:

1. The authors were modeling temporal changes of the text data using LDA from the standard Gensim Library. Authors should use dynamic topic model instead to capture the changes of topics over time.

2. Evaluating only on the topic coherence in choosing the topic number is not sufficient because it only evaluates each topic separately not across topics. You can have coherent topics but the topics can all look the same. A proper metric is called topic quality that combines topic coherent and topic diversity (Mimno et al 2011).

3. Authors mention computational overhead caused by choosing more topics. This is possibly because they were using the full-batch LDA training on 3.5 million records in their data. LDA can be inferred by stochastic variational inference using minibatch of the records (also known as online LDA). This should be implemented in the Gensim Library.

4. I am confused on whether the model was trained on real data and then applied to the simulated data or trained and applied directly on the simulated data. It seems that the former is true. I would argument that the latter is more realistic because in real world application, you can’t train on text in the absence of the outbreak at least not in the same time interval.

5. Figure 2 and 3a are in very poor quality. A more compact display is needed. Otherwise, Figure 2 and 3a should be in supplementary.

Minor comments:

6. The flow and the format of the paper makes it sometimes hard to follow. In particular, in Results section, figure legends were inserted in the middle of the text with large margin around them (e.g., page 14).

7. The authors call the veterinary clinical narratives of dogs as electronic health records (EHR). I wonder to what extent this is the case. There is no standardized data such lab tests LOINC or RxNorm prescription.

Reference:

D. Mimno, H. M.Wallach, E. Talley, M. Leenders, and A. McCallum. 2011. Optimizing semantic coherence in topic models. In Conference on Empirical Methods in Natural Language Processing.

Reviewer #2: The objective of the study was well motivated as to the use of topic modelling to classify EHR into categories that would allow for surveillance. What was not well explained is the criteria and definition of disease outbreak. The point was made that the utility of labelling would be to facilitate detection of outbreaks. When in the development of the outbreak should the system be making the decision that an outbreak is present? If it waits until the outbreak is at its worst then obviously this would mean that there would be little utility for prevention or control. On lines 186-8, an outbreak is equated to an increase in gastroenteric MPC but that is a very vague definition and does not take into account many different sources of variation (dog show in town, new clinic hours, holidays, etc.). These sources of variation are what makes syndromic surveillance so difficult because of the problem of false positives.

It was also mentioned that there is a great deal of uncertainty in topic modelling with regards to negatives - this is indeed a problem and I did not clearly see any explanations for how this was mitigated in this study. The other large issue with natural language data sets is misspellings, short forms, etc. From my own experience with human hospital and veterinary clinic records points to a huge variation of spellings unless the doctor/vet is constrained or there is post-processing to correct spellings or expand abbreviations (which is only possible for a known set of abbreviation variations). I did not see where it was explained about the processing of the data from the EHR and since I am not familiar with the UK systems for this type of data collection I am confused as to whether this is actually an issue.

I was also unclear as to what the effect was if the EHR was describing more than one condition of interest. It is also unclear if the reason that this topic modelling works well is because GI has mainly 2 symptoms (vomiting and diarrhoea). It was briefly discussed with respect to the overlap between cardiac and respiratory symptoms and diagnoses but it does leave the reader waiting for a subsequent study that looks at this question.

I would caution against justifying the choice of topics (30) because of computational cost without putting this into perspective. In other words, what type of computational system was used and how long did it take to produce results.

I did find the use of unsupervised methods appropriate and useful for this task and think that this study is an excellent first step in a more comprehensive use of this technique.

6. PLOS authors have the option to publish the peer review history of their article (what does this mean?). If published, this will include your full peer review and any attached files.

Reviewer #1: No

Reviewer #2: **Yes: **Deborah Stacey

---

## [Author Response · Author response to Decision Letter 0]

18 Jul 2021

Academic editor comments

.....Response:Apologies for incorrect formatting, we have reviewed the document and hope we have corrected any deviations from the requested format

2. We noted in your submission details that a portion of your manuscript may have been presented or published elsewhere.

.....Response:Figure 6c is very similar to a figure in a peer reviewed publication (Radford et al 2021). In the current paper, this panel shows the trajectory of a disease outbreak which is described in the Radford et al paper. It is used as a comparison to the signal detected by topic modelling which is under discussion in the current paper. The Radford paper is primarily reporting the disease outbreak. In the current paper, we are illustrating how a novel method of automated record annotation could have mirrored the outbreak report in the Radford et al paper. We are not stating that we have identified the outbreak using the included figure, rather, that its inclusion clearly illustrates the utility of topic modelling to perform the same task.

.....Response: SAVSNET data is collected in as anonymised form as possible, tabulated owner and pet names are not collected, neither are tabulated addresses or telephone numbers. However, the clinical free text data may include versions of these information. SAVSNET has automated redaction of such data to create deidentified text in the vast majority of cases, however, no deidentification software is 100% and some identifiers may get past this step, particularly unusual pet names which might identify the owner and as such the free text narratives are considered sensitive and are only shared through a data application process that commits users to appropriate curation of any data shared. The data access process is straightforward and can be initiated through e-mail approach to savsnet@liverpool.ac.uk or via our web-based data-access portal (https://savsnetvet.liverpool.ac.uk/DAPP). Further details of the data user agreement completed though this portal can be supplied if required for review.

 [ADR, PJN, GN, CA No funder reference number, DogsTrust, https://www.dogstrust.org.uk

ADR, PJN, GN BB/N019547/1, Biotechnology and Biological Sciences Research Council, https://bbsrc.ukri.org

ADR, PJN No funder reference number, British Small Animal Veterinary Association, https://www.bsava.com

The funders had no role in study design, data collection and analysis, decision to publish, or preparation of the manuscript.]

.....Response:We have removed mentions of funders from the Acknowledgments section and the funding statement remains the same

Additionally, because some of your funding information pertains to [commercial funding//patents], we ask you to provide an updated Competing Interests statement, declaring all sources of commercial funding.

In your Competing Interests statement, please confirm that your commercial funding does not alter your adherence to PLOS ONE Editorial policies and criteria by including the following statement: "This does not alter our adherence to PLOS ONE policies on sharing data and materials.” as detailed online in our guide for authors http://journals.plos.org/plosone/s/competing-interests. If this statement is not true and your adherence to PLOS policies on sharing data and materials is altered, please explain how.

Competing Interests statement

.....Response: We have not received commercial funding for this project. we presume the concern is that data contributors include commercial companies. In this case the companies include VetSolutions and Teleos whose practice management systems are modified to allow SAVSNET data collection and CVS who are corporate group of vet-practices from whom we collect data. None of these groups provide funding for the project or direct trajectory of research or publication and as such we have removed the word ‘support’ from the acknowledgement section. Please let us know if this requires further clarification.

Please include the updated Competing Interests Statement and Funding Statement in your cover letter. We will change the online submission form on your behalf.

.....Response:Figure 2 and Figure 3a have been converted to supporting information as advised by reviewer#1 and along with captions fro these captions for supporting information data-files have been appended to the manuscript.

Reviewers Comments -Reviewer #1:

The authors used topic model to mine text data to mine veterinary clinical narratives. They find that one of the topic mixture probabilities (topic 17) over time are indicative of gastroenteric presenting complaint (MPC). Using Bayesian binomial linear model, they confirm that the finding is statistically significant. The study show that it is possible to use topic model to detect outbreak of novel disease. Although there is no methodological innovations in topic modeling and the analyses are standard topic model analysis, it is a novel application and an interesting angle that the authors took in looking into the outbreak using topic models.

Major comments:

1. The authors were modeling temporal changes of the text data using LDA from the standard Gensim Library. Authors should use dynamic topic model instead to capture the changes of topics over time. 

.....Response:The main aim of our paper was to explore the feasibility of using topic modelling to identify a potential outbreak. For that, and as a first step, static models were considered a reasonable approach. Indeed, the signal identified using the ‘static’ model is demonstrated to be useful by its close mapping to the other outbreak metric (MPC). Still, following the reviewer’s comment, we have added to the paper how a dynamic topic model can be used to allow observers to infer potential evolution of the content of a given topic over time. We are currently evaluating a variety of other dynamic topic modelling approaches and topic modelling using tools such as BERTopic and Tomotopy, but we thought that a systematic evaluation of these is outside the scope of the current manuscript.

2. Evaluating only on the topic coherence in choosing the topic number is not sufficient because it only evaluates each topic separately not across topics. You can have coherent topics but the topics can all look the same. A proper metric is called topic quality that combines topic coherent and topic diversity (Mimno et al 2011). 

.....Response:This is a fair comment. However, for the purpose of monitoring for emergence of a syndrome signalling an outbreak, review of the topic contents would quickly identify ‘vacuous’ topics (we have added mention of these to the manuscript) meaning that the model can err on the side of too many topics while still being useful (as it was in this case). 

3. Authors mention computational overhead caused by choosing more topics. This is possibly because they were using the full-batch LDA training on 3.5 million records in their data. LDA can be inferred by stochastic variational inference using minibatch of the records (also known as online LDA). This should be implemented in the Gensim Library.

.....Response:We have opted to use a smaller subset to experiment with the number of topics, which is now hopefully clearly explained in the manuscript. Computational overheads were not the main limiting factor in building the model given the number of topics.

4. I am confused on whether the model was trained on real data and then applied to the simulated data or trained and applied directly on the simulated data. It seems that the former is true. I would argument that the latter is more realistic because in real world application, you can’t train on text in the absence of the outbreak at least not in the same time interval. 

.....Response:This is now clarified in the manuscript.

5. Figure 2 and 3a are in very poor quality. A more compact display is needed. Otherwise, Figure 2 and 3a should be in supplementary. 

.....Response:Figure 2 has been removed with explanation in the text. Figure 3a has been replaced with a figure illustrating clustering of topics with MPCs which illustrates the topic-mpc relationship more helpfully

Minor comments:

6. The flow and the format of the paper makes it sometimes hard to follow. In particular, in Results section, figure legends were inserted in the middle of the text with large margin around them (e.g., page 14). 

.....Response:The authors accept that the placement of figure legends does disrupt the text but figure legends were placed in accordance with editorial guidelines(https://journals.plos.org/plosone/s/figures): 

“Place figure captions in the manuscript text in read order, immediately following the paragraph where the figure is first cited. Do not include captions as part of the figure files or submit them in a separate document.”

7. The authors call the veterinary clinical narratives of dogs as electronic health records (EHR). I wonder to what extent this is the case. There is no standardized data such lab tests LOINC or RxNorm prescription. 

.....Response: We indeed have access to EHR (or EPR), which contain some structured data (species, breed, age, sex), prescription data and clinical narrative. While the data comes from various providers and is not standardized to any particular standard, it still does represent EHR. However, we have made it clearer in text that we only use the narrative part.

Reviewer comments - Reviewer #2: 

The objective of the study was well motivated as to the use of topic modelling to classify EHR into categories that would allow for surveillance. What was not well explained is the criteria and definition of disease outbreak. The point was made that the utility of labelling would be to facilitate detection of outbreaks. When in the development of the outbreak should the system be making the decision that an outbreak is present? If it waits until the outbreak is at its worst then obviously this would mean that there would be little utility for prevention or control. On lines 186-8, an outbreak is equated to an increase in gastroenteric MPC but that is a very vague definition and does not take into account many different sources of variation (dog show in town, new clinic hours, holidays, etc.). These sources of variation are what makes syndromic surveillance so difficult because of the problem of false positives.

.....Response:This paper seeks to illustrate that an unsupervised method can produce results comparable to supervised annotation of clinical records in the context of a disease outbreak. The exact mechanism for alerting the presence of an outbreak from this data remains to be decided and is being researched by the SAVSNET team. We have added a paragraph to the discussion alluding to this citing an elegant publication that offers a means to address this.

It was also mentioned that there is a great deal of uncertainty in topic modelling with regards to negatives - this is indeed a problem and I did not clearly see any explanations for how this was mitigated in this study. 

.....Response:In a sense the problem of negation is ignored in the face of a signal that tracks gastroenteric disease regardless of how the topic model emerges from the data. We do not address this in the discussion because the topic model does allow identification of the outbreak regardless of how negation of those words occurs in the original texts

The other large issue with natural language data sets is misspellings, short forms, etc. From my own experience with human hospital and veterinary clinic records points to a huge variation of spellings unless the doctor/vet is constrained or there is post-processing to correct spellings or expand abbreviations (which is only possible for a known set of abbreviation variations). I did not see where it was explained about the processing of the data from the EHR and since I am not familiar with the UK systems for this type of data collection I am confused as to whether this is actually an issue.

.....Response:The authors fully agree with this sentiment (having mis-spelled and abbreviated many a clinical narrative) but in effect, the topic modelling deals with this by treating all words/abbreviations/acronyms as tokens which comprise a topic agnostic of what those are. We have added detail to how tokens were generated in the methods.

I was also unclear as to what the effect was if the EHR was describing more than one condition of interest. It is also unclear if the reason that this topic modelling works well is because GI has mainly 2 symptoms (vomiting and diarrhoea). 

.....Response:Most documents were given a greater than zero probability for more than one topic and this vector of topic probabilities will be studied in further work. For the purpose of this paper, the authors felt that the key message that an unsupervised annotation method could identify an outbreak scenario which succeeded while only looking at a single topic annotation using the topic for which a narrative had a >0.5 probability. We have added a note in the discussion regarding further, more comprehensive analysis of narrative-topic probabilities.

It was briefly discussed with respect to the overlap between cardiac and respiratory symptoms and diagnoses but it does leave the reader waiting for a subsequent study that looks at this question.

.....Response:That study is on the way alongside use of newer topic modelling techniques that articulate word-meaning more effectively and therefore create more meaningful topics

I would caution against justifying the choice of topics (30) because of computational cost without putting this into perspective. In other words, what type of computational system was used and how long did it take to produce results.

.....Response:We have added detail regarding the modest computational capacity used for this study and hope that this spurs other investigators to consider this type of work even when high performance computing is not available.

I did find the use of unsupervised methods appropriate and useful for this task and think that this study is an excellent first step in a more comprehensive use of this technique.

.....Response:Many thanks, we feel this is very much a first step and, hopefully, one of many

---

## [Decision Letter · Decision Letter 1]

10 Nov 2021

Using topic modelling for unsupervised annotation of electronic health records to identify an outbreak of disease in UK dogs.

PONE-D-21-01941R1

Dear Dr. Noble,

We’re pleased to inform you that your manuscript has been judged scientifically suitable for publication and will be formally accepted for publication once it meets all outstanding technical requirements.

Kind regards,

Fernanda Dórea

Academic Editor

PLOS ONE

Additional Editor Comments:

I believe that this was enough to be able to track what has changed and the additions made, although as pointed out by one of the reviewers, in the future you may want to specify clearly the lines/sections where additions were made, and preferentially quote them in the response letter. Based on the acceptance recommendation from reviewer 2, plus my own judgement of how satisfactorily you answered the comments from reviewer 1, I have recommended acceptance of the paper. Although some comments remain, I deemed that another round of review would not likely cause further improvements in the current methods. I do urge you to read thoroughly through the reviewer 1 comments, whose pointers on methodological aspects of the papers can be an important consideration for future work. But I do agree with authors that for the purpose of early disease detection, the methods presented and the conclusions are sound, and therefore I recommended acceptance.

Reviewers' comments:

1.     We are all busy. It is difficult to find the revised content when the author simply asked that “it’s revised somewhere in the manuscript and go find it yourself”. I will appreciate if the authors write the revised part directly in their response to my comments AND indicate the page number where those revised parts appear in revised main text. Only this way can I properly and efficiently assess whether they have satisfactorily addressed my comments. This pertains to all my original comments. Please also do this in the next round of revision.

2.     In answering my comment 2 about evaluating topic quality as the product of topic diversity and topic coherence rather than just topic coherence, authors did not give satisfactory answer. Please perform topic quality analysis.

3.     The authors didn’t understand my comment 3. They said that they chose small subset of data to experiment with the number of topics, which IS ultimately due to the inefficient computational algorithm of the LDA they used because otherwise they don’t need to subset the data to experiment topic numbers. Therefore, I suggested using the online LDA to run on all their data not just subset.

4.     For my comment 6, typically you put figures and figure legends on top or bottom of the page not in the middle of the page.

5.     The revised pdf is merged with the PDF of the original submission. I only find figures from the original submission not the revised figures.

---

## [Editor Report · Acceptance letter]

1 Dec 2021

PONE-D-21-01941R1 

Using topic modelling for unsupervised annotation of electronic health records to identify an outbreak of disease in UK dogs. 

Dear Dr. Noble:

I'm pleased to inform you that your manuscript has been deemed suitable for publication in PLOS ONE. Congratulations! Your manuscript is now with our production department. 

Kind regards, 

on behalf of

Dr. Fernanda C. Dórea 

Academic Editor

PLOS ONE